# Genetic analyses identify pleiotropy and causality for blood proteins and highlight Wnt/β-catenin signalling in migraine

Hamzeh M. Tanha ●[1✉], The International Headache Genetics Consortium* & Dale R. Nyholt ●[1✉]

Migraine is a common complex disorder with a significant polygenic SNP heritability ($h^2_{SNP}$). Here we utilise genome-wide association study (GWAS) summary statistics to study pleiotropy between blood proteins and migraine under the polygenic model. We estimate $h^2_{SNP}$ for 4625 blood protein GWASs and identify 325 unique proteins with a significant $h^2_{SNP}$ for use in subsequent genetic analyses. Pleiotropy analyses link 58 blood proteins to migraine risk at genome-wide, gene and/or single-nucleotide polymorphism levels—suggesting shared genetic influences or causal relationships. Notably, the identified proteins are largely distinct from migraine GWAS loci. We show that higher levels of DKK1 and PDGFB, and lower levels of FARS2, GSTA4 and CHIC2 proteins have a significant causal effect on migraine. The risk-increasing effect of DKK1 is particularly interesting—indicating a role for downregulation of β-catenin-dependent Wnt signalling in migraine risk, suggesting Wnt activators that restore Wnt/β-catenin signalling in brain could represent therapeutic tools against migraine.

[1] School of Biomedical Sciences, Faculty of Health, and Centre for Genomics and Personalised Health, Queensland University of Technology (QUT), Brisbane, QLD, Australia. *A List of authors and their affiliations appear at the end of the paper. ✉email: hamzeh.mesriantanha@hdr.qut.edu.au; d.nyholt@qut.edu.au

**B**lood proteins are secreted from multiple tissues and cell types, commonly used in clinical settings. An alteration of a blood protein level is associated with disease via a shared biological mechanism (where a common biological process underlies the disease and the level of the protein) or a causal cascade (where the disease is caused by the aberrant protein level or vice versa); hence, blood proteins are promising as diagnostic biomarkers and therapeutic targets[1,2]. Genetics contributes to alterations in blood protein levels by direct influences on the production and secretion process and/or indirect influences through physiology and behaviour[3].

Genome-wide association study (GWAS) coupled with targeted proteomic methods have identified single-nucleotide polymorphisms (SNPs) associated with blood proteome levels, known as protein quantitative trait loci (pQTLs), including *cis-* and *trans-*pQTLs[1]. Many pQTLs colocalise with disease susceptibility loci, providing insight into the molecular function of the disease-associated loci; thus, functionally linking genetics to disease endpoints. Also, pQTLs have been successfully used as genetic instruments in Mendelian randomisation studies to identify causal relationships between protein levels and disease; and therefore, nominate novel drug targets for disease[1,2]. Although the blood levels of many proteins can be influenced by polygenic effects, studies of shared polygenic genetic architecture between blood proteome and common complex diseases (mostly polygenic), such as migraine, are limited in the literature.

Migraine is a highly prevalent neurological disorder of episodic severe headache imposing an enormous personal and socio-economic burden[4,5]. The pathobiology of migraine is poorly characterised, and the current treatment provides inadequate pain relief[6]. The study of the blood proteome in migraine may improve our understanding of migraine as a complex disease and identify novel therapeutic targets. In the present study, we utilised GWAS summary statistics data and statistical genetic approaches to investigate shared genetic influences (known as horizontal pleiotropy) and causality (known as vertical pleiotropy) between blood proteome levels and migraine risk.

We collected 4625 publicly available GWAS summary statistics for blood proteins and explored the extent of pleiotropy between migraine and the 325 blood proteins with a significant polygenic SNP heritability at three levels, including genome-wide, 18,236 protein-coding genes and SNPs within 1703 LD-independent loci. We also infer causality between migraine and the identified proteins under the polygenic model. Our findings link migraine to several proteins and their corresponding genes, that are mostly distinct from genome-wide significant migraine GWAS risk loci.

## Results

**Blood proteins with a significant polygenic signal.** Figure 1 illustrates the study's workflow. We collected 4625 publicly available GWAS summary statistics for blood proteins from six studies[1,2,7–10]. The median sample size across all collected protein GWASs was 3301 individuals. We first estimated the polygenic SNP heritability ($h^2_{SNP}$) for all 4625 blood protein GWASs using linkage disequilibrium score regression (LDSC) (Supplementary Data 1)[11]. We observed that $Z$ scores of $h^2_{SNP}\left(Z_{h^2_{SNP}}\right)$ are correlated with sample size ($r = 0.30$, 95% confidence interval [CI] = 0.27–0.32, $P < 2 \times 10^{-16}$), but not the number of included SNPs in LDSC analysis ($P = 0.87$). Conversely, we recently showed that estimated $Z_{h^2_{SNP}}$ for 972 blood metabolites is significantly associated with the number of included SNPs[12], suggesting a lower (genome-wide) polygenicity for blood proteome than blood metabolome. This might highlight that a larger fraction of blood proteome heritability is attributable to local signals

when the association is either at or near the gene locus that encodes the tested protein (*cis-*pQTLs) or is far from the encoding gene (*trans-*pQTLs). However, in the present study, we aimed to study only blood protein GWASs with a significant polygenicity (SNP-based heritability) that comprise genome-wide association signals, for example, from *cis-* and multiple *trans-*pQTLs effects. Indeed, to have the sufficient power to detect significant polygenic genetic overlap with the migraine GWAS, the individual blood protein GWAS must have a significant $h^2_{SNP}$. Therefore, we limited our analyses to GWASs with a nominally significant $h^2_{SNP}$ ($Z_{h^2_{SNP}} > 1.64$; $P_{h^2_{SNP}} < 0.05$). Also, as $h^2_{SNP}$ above one is not meaningful (and likely due to small sample size noise), we included GWASs with $0 < h^2_{SNP} < 1$ resulting in 362 blood protein GWASs (362/4625 = 7.8%; $4 \times 10^{-9} < P_{h^2_{SNP}} < 0.05$). Compared to our study of blood metabolome where 41.67% (405/972) of metabolites showed significant $h^2_{SNP}$[12], this finding also suggests that blood protein levels tend to be more attributable to local heritability than polygenicity (many *trans-*pQTLs spread across the genome).

For duplicated protein GWASs, the GWAS with the largest $Z_{h^2_{SNP}}$ was retained for further analyses, resulting in 325 unique proteins from five studies[1,2,8–10]. A summary of estimated $h^2_{SNP}$ for these 325 blood proteins is provided in Table 1. For migraine, the GWAS summary statistics from the IHGC 2016 study[13] with a liability scale $h^2_{SNP}$ of 10.35% (standard error [SE] = 0.51%, $Z_{h^2_{SNP}} = 20.29$, population prevalence = 0.15), was analysed.

**Genome-wide Pleiotropic effects influencing migraine and blood protein levels.** To investigate genome-wide pleiotropic effects between migraine and blood proteins, we implemented an approach similar to the SNP effect concordant analysis (SECA) approach[14]. Briefly, we first extracted LD-independent SNPs from migraine GWAS via *P*-value informed LD-clumping resulting in 113,251 LD-independent SNPs with the most significant migraine GWAS *P*-values. We next estimated Pearson correlation coefficients ($r$) between 113,251 SNP $Z$ scores from the migraine GWAS and $Z$ scores for the same set of SNPs from blood protein GWASs aligned to the same effect allele. This analysis was limited to 270 blood protein GWASs with > 60% overlap with the LD-independent migraine SNPs. Full details of this analysis are provided in the Methods section. All correlation results are provided in Supplementary Data 2. Of the included 270 correlation tests, we found SNP effects for blood levels of 13 proteins (13/270 = 4.81%) significantly correlated with SNP effects for increased migraine risk at FDR ≤ 0.05 (Fig. 2a).

Increased genetic risk for migraine was correlated with genetic factors for increased levels of six blood proteins (Fig. 2a), including matrix metallopeptidase 7 (MMP7, $r = 0.0141$, $P = 2 \times 10^{-6}$), CUB and zona pellucida like domains 1 (CUZD1, $r = 0.0114$, $P = 1 \times 10^{-4}$), KIT ligand (KITLG, $r = 0.0094$, $P = 2 \times 10^{-3}$), vascular endothelial growth factor A (VEGFA, $r = 0.0090$, $P = 3 \times 10^{-3}$), haemoglobin subunit theta 1 (HBQ1, $r = 0.0089$, $P = 3 \times 10^{-3}$), and growth differentiation factor 15 (GDF15, $r = 0.0087$, $P = 4 \times 10^{-3}$).

Additionally, increased genetic risk for migraine was correlated with genetic factors for decreased levels of seven blood proteins, including phenylalanyl-tRNA synthetase 2, mitochondrial (FARS2, $r = -0.0105$, $P = 4 \times 10^{-4}$), potassium voltage-gated channel subfamily E regulatory subunit 2 (KCNE2, $r = -0.0103$, $P = 6 \times 10^{-4}$), glutathione S-transferase alpha 4 (GSTA4, $r = -0.0095$, $P = 2 \times 10^{-3}$), carbonic anhydrase 10 (CA10, $r = -0.0092$, $P = 2 \times 10^{-3}$), inducible T cell costimulator (ICOS, $r = -0.0090$, $P = 3 \times 10^{-3}$), interleukin 19 (IL19, $r = -0.0089$, $P = 3 \times 10^{-3}$), and cysteine rich hydrophobic domain 2 (CHIC2, $r = -0.0088$, $P = 3 \times 10^{-3}$).

**a** **Pre–analysis procedure**
- Sourcing 4,625 GWASs for blood proteins from 6 separate studies
- Estimating polygenic SNP heritability ($h^2_{SNP}$) by LDSC
- Limiting further analyses to 325 GWASs with significant $h^2_{SNP}$ (Z score > 1.64) for unique blood proteins
- Sourcing migraine GWAS from the 2016 report of the IHGC
- Imputing HapMap3 missing SNPs by robust and accurate imputation from summary statistics (RAISS)

**b** **Extent of pleiotropy between migraine and 325 blood proteins**

**1) Pleiotropy at genome–wide**
- Global pleiotropic effects between migraine and blood proteins
- Method: Pearson correlation ($r$) between LD–independent SNP effects (Z scores)

**2) Pleiotropy at 18,236 protein–coding genes**
- Pleiotropic genes influencing migraine and blood proteins
- Method: Multi–marker analysis of genomic annotation (MAGMA) and Exact Binomial test

**3) Pleiotropy at SNPs within 1,703 LD–independent loci**
- Pleiotropic loci influencing migraine and blood proteins via a SNP
- Method: Pairwise GWAS (GWAS–PW)

**c** **Genetic correlation and causality between identified blood proteins and migraine**

**1) Bivariate genetic correlation**
- Significant genetic correlations were included in the causality analysis
- Method: LD score regression (LDSC) genetic correlation ($r_g$)

**2) Causal relationship**
- Genetic causality proportion (GCP) between blood proteins and migraine
- Method: Latent causal variable (LCV) model

**Fig. 1 Study workflow. a** We first collected 4625 publicly available GWAS summary statistics for blood proteins from six studies published between March 2016 and October 2020. Those studies with a sample size < 1000 were not resourced. The median sample size across all 4625 collected protein GWASs was 3301 individuals. We limited our analyses to GWASs with a nominally significant polygenic $h^2_{SNP}$ ($P_{\text{one-sided}}$ [$P_{h^2_{SNP}}$] < 0.05), resulting in 325 unique proteins. This step was necessary as we aimed to investigate the shared "polygenic" genetic architecture between blood proteins and migraine. More importantly, this step minimises the multiple testing burden. For migraine, we used migraine GWAS summary statistics from the 2016 report of the International Headache Genetics Consortium (IHGC) comprising 59,674 migraine cases and 316,078 migraine-free controls. To have a consistent list of SNPs for all GWASs (325 protein GWASs and the migraine GWAS), we imputed HapMap3 SNPs using RAISS software. **b** We conducted comprehensive genetic analyses estimating pleiotropy between the included 325 proteins and migraine at three levels, [i] genome-wide (by estimating Pearson correlation between LD-independent SNP effects), [ii] 18,236 protein-coding genes (applying MAGMA and Binomial test), and [iii] SNPs within 1703 LD-independent loci (applying GWAS-PW). **c** The causal influence of the identified proteins (from panel **b**) was estimated by implementing a genome-wide approach (LCV model) when there is a nominally significant genetic correlation (estimated by LDSC) between the blood level of the protein and the risk of migraine.

**Table 1 Summary of the 325 unique blood protein GWASs (with a significant SNP heritability) analysed in this study.**

| GWAS source study | Number of proteins | Sample size | $h^2_{SNP}$ range | $h^2_{SNP}$ SE range | $h^2_{SNP}$ median | $h^2_{SNP}$ SE median |
|---|---|---|---|---|---|---|
| Folkersen et al. 2020[1] | 66 | 30,931 | 0.05-0.27 | 0.02-0.1 | 0.11 | 0.03 |
| Kettunen et al. 2016[10] | 4 | 24,925 | 0.06-0.11 | 0.02-0.03 | 0.1 | 0.03 |
| Folkersen et al. 2017[9] | 4 | 3394 | 0.41-0.49 | 0.22-0.27 | 0.46 | 0.25 |
| Sun et al. 2018[2] | 200 | 3301 | 0.2-0.74 | 0.12-0.38 | 0.3 | 0.14 |
| Suhre et al. 2017[8] | 51 | 1335 | 0.68-0.98 | 0.37-0.58 | 0.83 | 0.44 |

$h^2_{SNP}$ SNP heritability, SE Standard error of SNP heritability.

**Pleiotropic genes influencing migraine and blood protein levels.** We used MAGMA gene-based analysis;[15] first, to assign SNPs to 18,236 protein-coding genes; second, to estimate the association $P$-value for each gene ($P_{\text{gene}}$), for migraine and the included 325 blood proteins. Migraine- and protein-associated genes were defined at $P_{\text{gene}} \leq 0.05/18,236$. To study the presence of pleiotropy between migraine and blood protein levels at the gene level, we tested if genes associated with migraine risk are statistically enriched in the genes associated with blood protein levels using the *Exact Binomial Test*. The alternative hypothesis was that the observed proportion (the number of genes associated with both migraine and the protein divided by the number of genes associated with the protein) is greater than the null proportion (the number of genes associated with migraine [494] divided by the number of all genes [18,236]). More details are provided in the Methods section. This analysis revealed that

migraine-associated genes are enriched in the genes influencing blood levels of 15 proteins at FDR ≤ 0.05 (Fig. 2b). We recently suggested that a significant enrichment of associated genes (biologically meaningful units) for two traits, here migraine and a blood protein level, provides insights into a shared biological mechanism underlying two traits known as biological or horizontal pleiotropy[16]. The top five identified proteins having a pleiotropic effect at the gene level with migraine are cathepsin S (CTSS, $P_{binomial-test} = 2 \times 10^{-29}$), chromodomain Y-linked 1 (CDY1, $P_{binomial-test} = 2 \times 10^{-21}$), follistatin (FST, $P_{binomial-test} = 2 \times 10^{-21}$), matrix metallopeptidase 1 (MMP1, $P_{binomial-test} = 2 \times 10^{-21}$), RB binding protein 5, histone lysine methyltransferase complex subunit (RBBP5, $P_{binomial-test} = 4 \times 10^{-21}$).

Our original MAGMA result from migraine GWAS suggested 494 genes associated with migraine risk at $P_{\text{gene}} \leq 0.05/18,236$ (Supplementary Fig. 1a and Supplementary Data 3). To identify

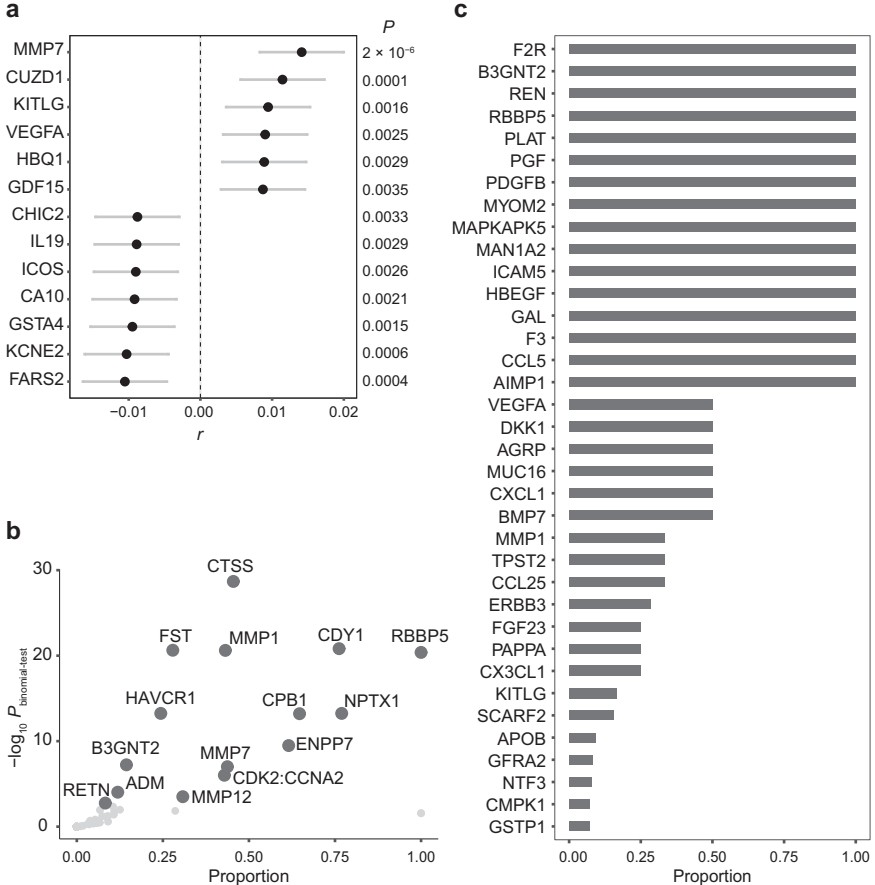

**Fig. 2 Extent of pleiotropy between migraine and blood proteins. a** Pleiotropy at genome-wide. Pearson correlation (two-sided) between Z scores ($r$) of 113,251 LD-independent SNPs from migraine GWAS and from blood protein GWASs produce significant results for 13 proteins after adjusting for multiple testing (FDR ≤ 0.05). All correlation results are provided in Supplementary Data 2. Circles represent the correlations ($r$) and the error bars indicate the 95% confidence interval. Unadjusted $P$-values for the correlations are shown on the right. **b** Pleiotropy at 18,236 protein-coding genes identified by using MAGMA results and the *Exact Binomial test* (one-sided). Pleiotropic genes significantly influence migraine and blood levels of 15 proteins after adjusting for multiple testing (FDR ≤ 0.05). The proportion for each protein is the number of genes associated with both migraine and the protein divided by the number of genes associated with the protein. **c** Pleiotropy at SNPs within 1703 LD-independent loci identified by GWAS-PW. Pleiotropic SNPs influence migraine and blood levels of 36 proteins (posterior probability ≥ 0.9). The proportion of the LD-independent loci that influence the blood level of the protein also influence migraine via a shared SNP ranged between 1 and 0.07 (see Methods section for details). Full GWAS-PW results are available in Supplementary Data 5 and 6. For consistency, proteins are reported by encoding gene symbols throughout the paper.

genes that contribute to both migraine susceptibility and blood protein levels, we combined MAGMA results ($P_{gene}$ values) from migraine GWAS and the 15 identified proteins (Fig. 2b). This combined analysis identifies genes that are likely contributing to both migraine risk and alterations in blood levels of the 15 proteins. Also, it may yield novel genes associated with migraine risk by increasing statistical power. Full details are provided in the Methods section. This combined analysis identified 651 genes at *Combined-$P_{gene}$* ≤ 0.05/291,776 (291,776 is calculated based on the number of tests [16 × 18,236]) of which 254 genes are novel (Supplementary Fig. 1b and Supplementary Data 4).

**Pleiotropic SNPs influencing migraine and blood protein levels.** We used the pairwise GWAS (GWAS-PW) approach to identify an independent genomic locus that influences both migraine and the blood level of a protein via a shared SNP[17]. GWAS-PW estimates four posterior probabilities for up to 1703 LD-independent loci supporting four scenarios; first, the genomic locus only affects migraine; second, the genomic locus only affects the blood protein level; third, the genomic locus affects both migraine and the blood protein level via a shared SNP; fourth, the genomic locus affects both migraine and the blood protein level via two

separate SNPs. We defined a locus as the pleiotropic associated SNP when its posterior probability supporting the third scenario (PPA3) is ≥ 0.9. GWAS-PW findings including the number of loci that fit each of the four scenarios for the included 325 proteins are available in Supplementary Data 5.

GWAS-PW identified 24 pleiotropic loci across 12 chromosomes influencing migraine and one or more blood proteins via shared SNPs (Supplementary Data 6)—connecting migraine to 36 blood proteins. The proportion of the LD-independent loci that influence the blood protein level also influence migraine via shared SNPs ranged between 1 to 0.07 (Fig. 2c). Among the identified 36 blood proteins, five proteins have two or more pleiotropic loci with migraine, including four loci for erb-b2 receptor tyrosine kinase 3 (ERBB3), three loci for coagulation factor II thrombin receptor (F2R), and two loci for UDP-GlcNAc:betaGal beta-1,3-N-acetylglucosami-nyltransferase 2 (B3GNT2), vascular endothelial growth factor A (VEGFA) and scavenger receptor class F member 2 (SCARF2).

The top three pleiotropic loci that influence migraine and blood proteins (Fig. 3) are chromosome 9: 135.3–137 Mb (lead SNP rs495828; associated with migraine risk and levels of 11 blood proteins), chromosome 10: 63.3–65.8 Mb (lead SNP rs10761741; associated with migraine risk and levels of 7 blood

proteins) and chromosome 10: 95.4–96.2 Mb (lead SNP rs10786156; associated with migraine risk and levels of 3 blood proteins). Regional Manhattan plots for these pleiotropic loci were generated using LocusZoom online tool (accessed Jun 2021, http://csg.sph.umich.edu/locuszoom)[18].

**Blood proteins with a causal effect on migraine**. Our cross-trait analyses provide an overview of pleiotropy between migraine and blood proteins at genome-wide, genes, and SNPs levels, identifying 58 unique blood proteins (out of 325 tested proteins) with pleiotropic effects on migraine (Fig. 2). Next, we tested for causal relationships between migraine and the 58 implicated blood proteins by applying the latent causal variable (LCV) model[19]. LCV tested whether the bivariate genetic correlation ($r_g$ estimated by LDSC)[20] between migraine and blood proteins is mediated by shared aetiology (horizontal pleiotropy) or causation (vertical pleiotropy). LCV analysis is not limited to a few genetic instruments; thus, it is more consistent with the polygenic model. As recommended by LCV developers[19], we limited our analysis to eight proteins whose blood levels have a significant genetic correlation ($P_{rg} < 0.05$) with migraine risk (Table 2). Genetic correlation results help to interpret LCV findings by providing causal effect size and the direction of the relationship (whether higher or lower blood levels of a protein increase the risk of the disease)[19].

The LCV approach identified that alterations in blood levels of five proteins have a significant genetic causality on migraine with genetic causality proportion (GCP) ranging from 0.37 to 0.88. It has been suggested that a causal relationship with the GCP ≥ 0.6 is extremely unlikely to be false-positive, suggesting a partial to full genetic causality. A genetic causality with GCP < 0.6 suggests low to medium (or even partial) genetic causality[19]. The LCV significant findings (FDR ≤ 0.05) include genetic causality on migraine for higher levels of two blood proteins, including dickkopf WNT signalling pathway inhibitor 1 (DKK1, GCP = 0.88, $P_{GCP} = 4 \times 10^{-48}$) and platelet derived growth factor subunit B (PDGFB, GCP = 0.70, $P_{GCP} = 2 \times 10^{-17}$), and lower levels of three blood proteins, including phenylalanyl-tRNA synthetase 2, mitochondrial (FARS2, GCP = 0.69, $P_{GCP} = 1 \times 10^{-8}$), glutathione S-transferase alpha 4 (GSTA4, GCP = 0.56, $P_{GCP} = 3 \times 10^{-9}$) and cysteine rich hydrophobic domain 2 (CHIC2, GCP = 0.37, $P_{GCP} = 3 \times 10^{-7}$).

**Biological characterisation of the findings**. Functional annotation of the findings was conducted using the g:Profiler web server[21]. The 325 unique proteins that were tested in this study are mostly located in the "extracellular region" (source: GO cellular component, FDR = $8 \times 10^{-61}$) as blood proteins are mostly secreted proteins from a variety of tissues; and enriched in the "immune system process" (source: GO biological process, FDR = $1 \times 10^{-27}$). All annotation results are available in Supplementary Data 7.

Our cross-trait genetic analyses found evidence for pleiotropy between migraine risk and alterations in blood levels of 58 proteins (Fig. 2), all encoded by genes in a wide variety of tissues and cell types. Interestingly, our genetic analyses of blood proteome highlighted a set of genes in migraine that have not been implicated by standard migraine GWAS. For example, MAGMA gene analysis utilising migraine GWAS failed to produce even nominally significant P-values for more than 60% of the 58 identified protein-encoding genes ($P_{gene} ≥ 0.05$ for 35 proteins and $P_{gene} < 0.05$ for only 23 proteins, Supplementary Data 8). Therefore, this study suggests that integrating GWAS from a complex trait such as migraine with blood proteome GWASs produces novel findings that are distinct from the complex trait genome-wide significant GWAS loci. The 58 identified proteins are mostly enriched in "positive regulation of

protein kinase B signalling" (source: GO biological process, FDR = $1 \times 10^{-6}$). All annotation results are available in Supplementary Data 9.

Furthermore, we used consensus normalised expression levels from the Human protein atlas (HPA) consensus dataset[22] to compare the RNA expression levels of the 58 identified proteins to those proteins that have no detected pleiotropy with migraine (~267). For more details see the Methods section and Supplementary Data 10. This analysis found that RNA expression levels of proteins that have pleiotropy with migraine are significantly lower than other tested proteins across all HPA tissues (mean of 1.15 *versus* 1.35, $P_{t-test} = 6 \times 10^{-8}$). This suggests that proteins with lower RNA expression levels have a higher probability to be associated with a complex disorder, such as migraine, probably due to their lower tolerance for any alteration. However, we identified no significant difference between HPA proteins expression levels[23] of the proteins that have pleiotropy with migraine compared to other tested proteins ($P_{t-test} = 0.85$) (Supplementary Data 11).

Lastly, our scan for inferring causality identified five blood proteins (DKK1, PDGFB, GSTA4, FARS2 and CHIC2) with a significant genetic causality on migraine. The encoding genes of these proteins are significantly enriched in the "Brain stem compression" phenotype with the overlap of *DKK1* and *PDGFB* (source: human phenotype ontology, FDR = 0.038). We also visually compared the RNA expression levels of *DKK1*, *PDGFB*, *GSTA4*, *FARS2* and *CHIC2* to the genes that encode the recent migraine therapeutic target, calcitonin gene-related peptide (*CGRP*)[6], including *CALCA* and *CALCB* (Supplementary Fig. 2) across ten HPA brain regions. We noted that the patterns of the RNA expression levels of *DKK1* and *CALCA* across brain tissues are highly similar, both are highly expressed in "Pons and medulla oblongata" (a part of brain stem) compared to other brain regions (Supplementary Fig. 2).

## Discussion

The present study aimed to identify blood proteins that have pleiotropy with the risk of migraine. Such a pleiotropy suggests an alteration of the protein level in migraine patients, indicating a shared genetic mechanism and/or a causal relationship. As migraine, like many other common complex traits, is highly polygenic, we designed our pipeline under the polygenic model by [i] limiting our analyses to those blood proteins that have a significant polygenic $h^2_{SNP}$, resulting in 325 proteins (/4625), [ii] exploring the extent of pleiotropy between blood levels of proteins and risk of migraine at three different levels of genome-wide, genes and SNPs within LD-independent loci, and [iii] inferring causal effects of the identified proteins on migraine using a genome-wide method. The study workflow is shown in Fig. 1. This pipeline can be used to study shared polygenic genetic architecture between blood proteome and any common trait of interest and can provide a more comprehensive overview of the complex relationship between proteins and disease than simply colocalising pQTLs with the disease risk loci (which is more consistent with a local heritability model).

Comparing the heritability results from the present study to our recent study of blood metabolome[12] suggests a lower polygenicity for blood proteome than blood metabolome. First, polygenic SNP heritability was not associated with the number of included SNPs for the 4625 blood proteins, whereas it was significantly associated in our study of 972 blood metabolites. Second, a smaller proportion of blood proteins showed significant polygenic $h^2_{SNP}$ compared to blood metabolites (7.8% *versus* 41.67%). Taken together, our comparisons suggest that blood metabolites appear more polygenic than blood proteins, which is

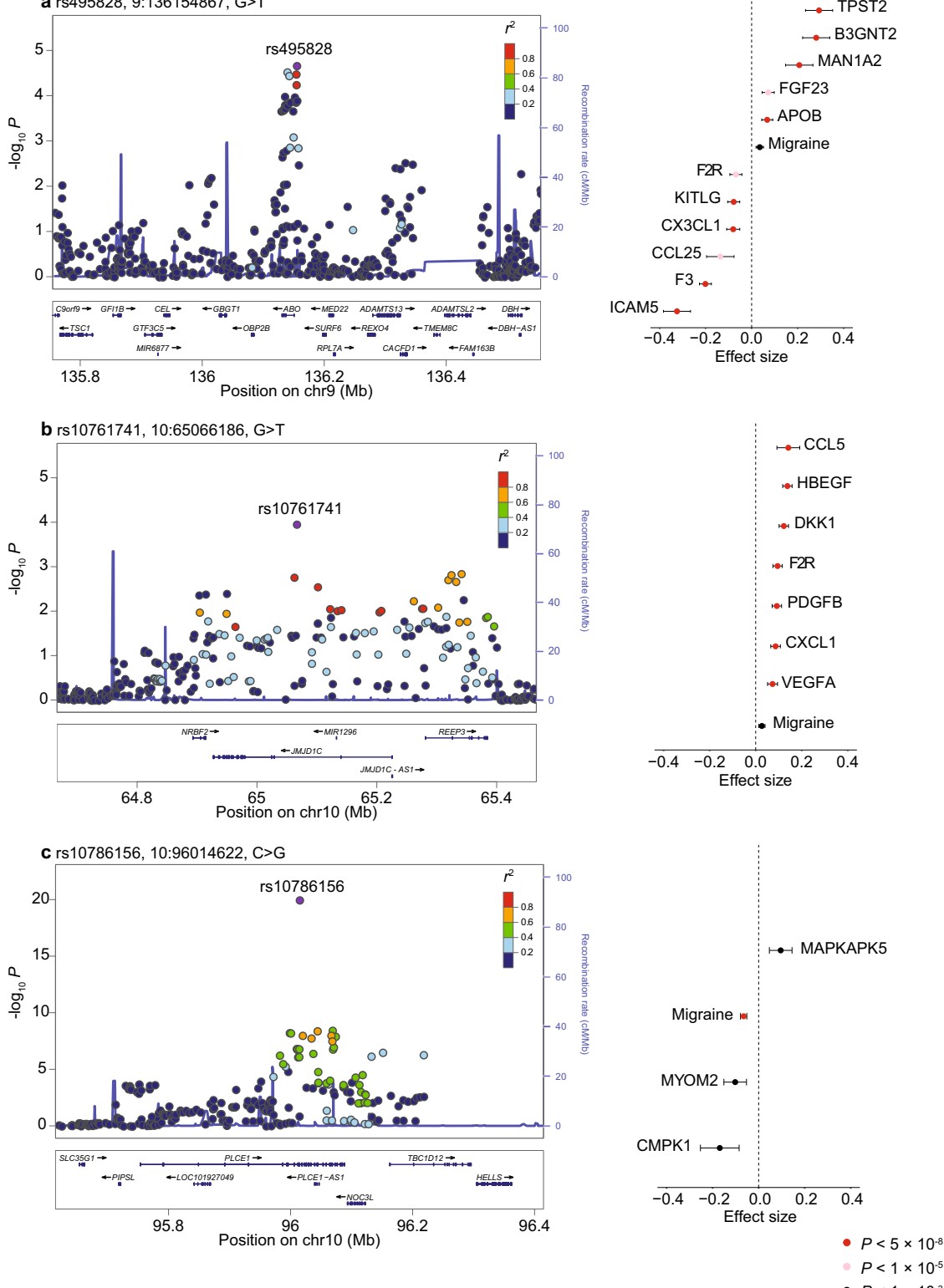

**Fig. 3 Top three genomic loci influencing migraine and 21 blood proteins via shared SNPs.** LocusZoom plots of migraine GWAS for three pleiotropic loci that affect migraine and multiple proteins via shared SNPs are shown on left. Association effect sizes and their 95% confidence intervals of the shared SNP in each locus for migraine and identified blood proteins are shown on right. Unadjusted *P*-values for the associations from the migraine and protein GWASs are shown in red ($P < 5 \times 10^{-8}$), pink ($P < 1 \times 10^{-5}$) and black ($P < 1 \times 10^{-3}$). **a** Chromosome 9: 135.3–137 Mb affects migraine and 11 blood proteins via rs495828 (position, 9:136154867; effect allele, T). **b** Chromosome 10: 63.3–65.8 Mb affects migraine and seven blood proteins via rs10761741 (position, 10:65066186; effect allele, T). **c** Chromosome 10: 95.4–96.2 Mb affects migraine and four blood proteins via rs10786156 (position, 10:96014622; effect allele, G). Note, proteins are reported by encoding gene symbols.

**Table 2 Causal relationship between blood proteins and migraine.**

| Protein | $r_{rg}$ | $SE_{rg}$ | $P_{rg}$ | GCP | $SE_{GCP}$ | $P_{GCP}$ |
|---|---|---|---|---|---|---|
| DKK1 | 0.11 | 0.05 | 0.021 | 0.88 | 0.08 | **$4 \times 10^{-48}$** |
| PDGFB | 0.15 | 0.06 | 0.007 | 0.7 | 0.18 | **$2 \times 10^{-17}$** |
| GSTA4 | −0.22 | 0.11 | 0.042 | 0.56 | 0.26 | **$3 \times 10^{-9}$** |
| FARS2 | −0.20 | 0.09 | 0.034 | 0.69 | 0.07 | **$1 \times 10^{-8}$** |
| CHIC2 | −0.14 | 0.05 | 0.008 | 0.37 | 0.13 | **$3 \times 10^{-7}$** |
| B3GNT2 | −0.15 | 0.07 | 0.046 | −0.39 | 0.38 | 0.346 |
| ICOS | −0.22 | 0.10 | 0.023 | −0.04 | 0.57 | 0.601 |
| MMP7 | 0.23 | 0.10 | 0.030 | −0.01 | 0.56 | 0.870 |

Here we limited causality inference, latent causal variable (LCV), to blood proteins with significant bivariate genetic correlation, LD score regression (LDSC) genetic correlation ($r_g$). Out of the identified 58 blood proteins with pleiotropic effects on migraine, 8 proteins have nominally significant $r_g$ with migraine (unadjusted $P_{rg} < 0.05$). GCP and $SE_{GCP}$ are posterior mean genetic causality proportion and posterior standard error estimated by the LCV model. Unadjusted $P_{GCP}$ tested the null hypothesis of no genetic causality. Significant causal effects of blood proteins on migraine risk after adjusting for multiple testing (FDR ≤ 0.05) are shown in bold. Note, proteins are reported by encoding gene symbols.

plausible given multiple proteins can play catalytic or regulatory roles in anabolism, catabolism, absorption, secretion and transportation of a metabolite[24].

Our scans for pleiotropy genetically link 13, 15 and 36 blood protein levels to migraine risk at genome-wide, genes and SNPs levels, respectively, implicating a total of 58 "unique" proteins in migraine. Only six proteins showed evidence for pleiotropy at multiple levels, including genome-wide and genes levels (MMP7), genome-wide and SNPs levels (KITLG and VEGFA), and genes and SNPs levels (MMP1, RBBP5 and B3GNT2). Therefore, as reported by us[12] and others[17,20], findings from pleiotropy analyses at different levels are mostly distinct and thus complementary. Our pleiotropy at the genome-wide level, a method similar to LDSC genetic correlation, provides a direction for the relationship (a signed correlation). This method requires widespread polygenic signals with a consistent direction of effects between migraine and a protein. Conversely, the pleiotropy at the genes and SNPs levels provides no direction for the relationships. The pleiotropy test at the gene level requires polygenic signals, but a shared monogenic signal is sufficient for a SNP to be pleiotropic between migraine and a protein. Hypothetically, a polygenic signal from a protein level means many trans-pQTLs are involved but a monogenic signal means only one genetic locus is involved, either a cis- or trans-pQTL.

The identified proteins with a pleiotropic effect on migraine were followed up by inferring causality under the polygenic model implementing the genome-wide LCV approach, which is not limited to a few genetic instruments[19]. The LCV analyses found a significant genetic causality on migraine for higher levels of two blood proteins, DKK1 and PDGFB. Our biological characterisation showed that DKK1 and PDGFB are involved in the "Brain stem compression" phenotype. We also noted that *DKK1* and *CALCA*—which encodes CGRP, a migraine-specific drug target—have a similar pattern of RNA expression across ten HPA brain regions with a high expression in "Pons and Medulla oblongata" that is a part of the brain stem. Interestingly, changes in the activity of the brain stem during migraine attacks have been suggested[6]. The protein encoded by *DKK1* is an endogenous antagonist of the Wnt signalling pathway that acts by binding to the low-density lipoprotein (LDL) Receptor Related Protein 6 (LRP6) co-receptor and inhibits beta-catenin-dependent Wnt signalling[25]. DKK1 has been introduced as a promising therapeutic target for the treatment of Alzheimer's disease (AD). An increased level of DKK1 in the brains of AD patients has been linked to the downregulation of Wnt signalling, and cerebral

amyloid angiopathy (CAA)[26,27]. Also, a Wnt signalling inhibitor increased neuropathic pain in a rat model[28]. Our finding of a strong causal effect of higher levels of DKK1 (GCP = 0.88) on migraine risk might be linked to the dysfunction of Wnt signalling and CAA. Interestingly, it has been shown that migraine is an early symptom of CAA[29] and a study of 679 community-dwelling seniors (65 + years) in Manitoba, Canada, found that participants with AD were over four times more likely (OR = 4.22; 95% CI = 1.59–10.42) to have a history of migraines[30]. Interestingly, SNPs within the LDL receptor related protein 1 (*LRP1*) gene were one of the first loci robustly associated with common migraine via GWAS[5,31] and LRP1 is also implicated in AD[32]. Therefore, as proposed for AD, Wnt activators that restore Wnt/β-catenin signalling in brain, particularly those targeting Wnt antagonist DKK1 and Wnt receptor LRP6, could represent novel therapeutic tools for migraine treatment. PDGFB, a member of the PDGF/VEGF growth factor family, can either homodimerise (PDGF-BB) or heterodimerise with PDGFA (PDGF-AB) to activate PDGF tyrosine kinase receptors. It has been shown that PDGF-BB activates nociceptive neurons via inhibiting a class of potassium channels; thus, mediates inflammatory pain in rats[33]. Mutations in *PDGFB* cause brain calcification disorders and; interestingly, it has been shown that migraine is commonly present in the mutation carriers[34,35]. Taken together, our current results suggest DKK1 and PDGFB as two promising therapeutic targets against migraine.

Additionally, our LCV analyses found a significant genetic causality on migraine for lower levels of three blood proteins, including FARS2, GSTA4 and CHIC2. FARS2, a member of mitochondrial protein translation system, transfers phenylalanine to its cognate transfer RNA (tRNA). Therefore, the causal effect of lower levels of FARS2 on migraine suggests higher levels of free phenylalanine in migraine patients. Indeed, we recently found a significant genome-wide genetic overlap between higher levels of blood phenylalanine and migraine risk[12]. The causal effect of lower levels of GSTA4, an antioxidant enzyme, on migraine is consistent with previously reported higher levels of oxidants (such as nitric oxide) and lower levels of antioxidants (such as glutathione and glutathione S-transferase) in migraine patients[12,36,37]. Hence, our findings suggest that the imbalance between production and detoxification of oxygen reactive species in migraine patients might be caused by lower levels of GSTA4. Lastly, although lower levels of CHIC2, a cysteine-rich hydrophobic (CHIC) protein localised to vesicular structures and the plasma membrane, have a comparatively lower genetic causality on migraine (GCP = 0.37) and the mechanism underlying this causal relationship is unclear in the literature, our results indicate functional follow-up of CHIC2 in migraine is warranted.

Our findings suggest that of the identified pleiotropic effects between blood levels of 53 proteins and risk of migraine, for those with non-significant LCV results, their relationship with migraine is more consistent with shared biology (horizontal pleiotropy) than causality. In the following, we discuss some of the interesting horizontal pleiotropy findings. Matrix metallopeptidases (MMPs) with at least 25 members are involved in the neuroinflammation[38] and permeability of the blood-brain barrier (BBB)[39]. MMPs in the brain are activated by the cortical spreading depression (CSD) that occurs during migraine attacks[40]. Also, SNPs near MMP encoding genes have been associated with migraine risk[13,41]. Similarly, higher levels of blood MMP9 were found in migraine patients compared to controls[40]. We identified a significant genome-wide pleiotropy between higher levels of MMP7 and migraine risk. The other identified pleiotropic effects that link MMPs to migraine risk are pleiotropy at the gene level for MMP1, MMP7 and MMP12, and at the SNPs level for MMP1. Notably, MMP9 that has been frequently linked

to migraine[39,40,42], was not included in our analyses as its $Z_{h^2_{SNP}} = 0.98$ ($P_{h^2_{SNP}} = 0.16$, Supplementary Data 1). Also, our combined gene-based analysis identified three migraine-associated *MMPs*, including *MMP7*, *MMP20* and *MMP27* (Supplementary Data 4). Moreover, it has been shown that BBB permeability and protein expression of *VEGFA* were increased in rats with induced recurrent headache[43]. VEGFA plays a role in inflammation and pain[44]. Likewise, we identified a genome-wide pleiotropy between higher levels of VEGFA and migraine risk. We also identified pleiotropy at the SNPs level between the blood levels of VEGFA and migraine risk. Although disruptions in BBB permeability have been hypothesised for migraine[45], increased permeability of BBB has only been found in rare types of migraine, including familial hemiplegic migraine[46] and status migrainosus[47]. Our findings of higher levels of MMP7 and VEGFA suggest enhanced permeability of BBB in all migraine patients.

There is interesting evidence in the literature that directly or indirectly links the identified proteins to migraine. We found a genome-wide pleiotropy between higher levels of GDF15 and migraine risk. Interestingly, elevated levels of GDF15 cause nausea and vomiting in rats[48] which are common symptoms of migraine[6]. Moreover, we found a genome-wide pleiotropy between lower levels of an accessory subunit of voltage-gated potassium channels, KCNE2, and migraine risk. Interestingly, potassium channels with a variety of biological functions (e.g., regulating neurotransmitter release) have been linked to migraine in genetic studies[49] and to migraine attack mechanisms[6]. We also found a genetic relationship between migraine and lower levels of CA10 blood protein that is mainly expressed in the central nervous system and its gene locus has been associated with multisite chronic pain[50]. Furthermore, we identified genome-wide pleiotropy between migraine and lower levels of IL19, an anti-inflammatory cytokine that belongs to the IL10 cytokine subfamily. This finding is in line with previously reported increased levels of pro-inflammatory and decreased levels of anti-inflammatory blood cytokines in migraine patients[51].

Among the identified 36 blood proteins with pleiotropic effects on migraine at SNPs within LD-independent loci, five proteins have two or more pleiotropic SNPs with migraine, including four SNPs for ERBB3, three SNPs for F2R, and two SNPs for B3GNT2, VEGFA and SCARF2. ERBB3 is a member of the epidermal growth factor (EGF) receptor family that binds to and is activated by neuregulin-1[52], an EGF protein that has been associated with neuropathic pain in rats[53]. F2R (also known as PAR1) is a transmembrane receptor that is associated with thrombotic response[54]. Interestingly, it has been found that migraine patients have an increased risk for venous thromboembolism[55]. It has been shown that the *B3GNT2* gene locus is associated with rheumatoid arthritis, a chronic inflammatory disorder[56]. It has been found that patients with rheumatoid arthritis have a higher risk for migraine and neuropathic pain[57]. Finally, SCARF2 which is involved in the degradation of acetylated low-density lipoprotein (LDL) might be linked to a higher level of LDL and its related metabolites in migraine patients[16].

Intriguingly, the identified migraine-associated blood proteins in this study are largely distinct from migraine GWAS findings (the coding genes of the identified proteins are far from GWAS hits). This is a notable finding, which we expect to be due to our study of polygenic pleiotropy between migraine and blood proteome where multiple non-coding GWAS loci and *trans*-pQTLs are involved, and where a protein expression level is influenced by distant genetic loci through complex indirect processes[3]. Given the discordance of RNA and protein levels for secreted proteins (i.e., blood proteins)[58], it is possible that the findings from blood transcriptome to be more consistent with GWAS findings than

proteome. The pleiotropy analysis between blood proteome and a disease can also identify disease-associated proteins that have their encoding genes located on sex chromosomes that may not be assessed via genotyping arrays—e.g., our finding of significant gene-level pleiotropy between blood levels of CDY1 (coded by a gene located on the Y chromosome) and risk of migraine. Overall, our findings indicate that our pipeline for the genetic study of blood proteome in common complex diseases has enormous potential to identify important and novel disease-associated proteins which a standard disease GWAS will miss.

The potential limitations of the present study are similar to the limitations we listed in our recent genetic study of metabolites in migraine[12]. First, because this study utilised GWAS data from European populations, our findings may not be generalisable to other ancestries. Second, while the unknown sample overlap between migraine GWAS and protein GWASs may result in slightly inflated findings, we do not expect this to have an impact on our conclusions. Third, this study explored genetic pleiotropy between migraine and proteins using methods that are less sensitive to protein GWAS sample size. These methods, however, have their own set of restrictions. For example, the Pearson correlations between LD-independent SNP effects do not estimate the magnitude of migraine-protein associations; rather, this approach finds proteins with genome-wide pleiotropic effects on migraine. Fourth, here we mainly aimed to study shared polygenic genetic effects; thus, only focused on the 7.8% (362/4625) of blood proteins with a significant polygenic (genome-wide) $h^2_{SNP}$ (estimated by LDSC). Therefore, the genetic overlaps between blood proteins and migraine under the local heritability model remain to be studied. Fifth, although blood proteins are secreted from multiple tissues, our results may not accurately reflect tissue-specific migraine-associated proteins. Finally, replication of our findings in larger migraine and protein GWASs, as well as validation in randomised clinical trials, will be beneficial.

The colocalisation of disease-associated genetic variants with pQTLs identifies intermediate proteins functionally linking genetics to disease endpoints. Although alterations in levels of many proteins are attributable to polygenicity where many genetic variants with small effects are involved, shared polygenic genetic architectures between proteome and complex traits have been rarely studied. Our systematic investigation of pleiotropy between migraine and blood levels of 325 unique proteins with significant polygenic SNP heritability, implicates blood level alterations of 53 proteins in migraine risk via shared genetic influences. In addition, we identified that higher levels of DKK1 and PDGFB, and lower levels of FARS2, GSTA4 and CHIC2 causally increase the risk of migraine. Future studies should examine whether altering the blood levels of these proteins reduces migraine occurrence in migraine patients.

## Methods

**GWAS summary statistics.** This study utilised GWAS summary statistics for blood levels of 4625 proteins from six studies published between March 2016 and October 2020[1,2,7–10]. Those studies with a sample size < 1000 were not included. Supplementary Table 1 includes information on the six studies' characteristics, population, demographics, and proteomics platforms. Variant information from the human genome build 37 (GRCh37 [hg19]) was utilised to annotate the GWAS summary statistics with missing information (e.g., rsIDs and non-effect alleles).

For migraine, we used the summary statistics from the 2016 International Headache Genetics Consortium (IHGC) GWAS comprising 59,674 migraine cases and 316,078 migraine-free controls[13]. This 2016 migraine GWAS contains association results for a total of 8,049,884 SNPs. All GWAS summary statistics in this study are from European populations.

**Estimation of SNP heritability and filtration.** Polygenic SNP heritability ($h^2_{SNP}$) for the 4625 blood proteins were estimated using linkage disequilibrium score regression (LDSC, https://data.broadinstitute.org/alkesgroup/LDSCORE/)[11]. We

used the European LD scores for HapMap3 SNPs calculated by LDSC developers utilising the European 1000 Genomes Project phase 3 (1000 G) as the LD reference panel[59,60]. When the GWAS summary statistics included the relevant information, SNPs with minor allele frequency (MAF) < 0.01 and imputation quality < 0.9 were removed. Supplementary Data 1 contains $h^2_{SNP}$ estimates for all 4625 blood proteins. We selected GWASs when the $Z$ score of $h^2_{SNP}$ ($Z_{h^2_{SNP}}$) is > 1.64 ($P_{one-sided}$ [$P_{h^2_{SNP}}$] < 0.05); also, as $h^2_{SNP} > 1$ is not meaningful, suggesting sample size noise, we included only GWASs with $0 < h^2_{SNP} < 1$ resulting in 362 blood protein GWASs. For duplicated GWAS proteins (e.g., two GWASs for the same protein), the GWAS with the highest $Z_{h^2_{SNP}}$ was selected. In total, 325 unique blood protein GWASs were included for further genetic analyses. For consistency, included proteins are reported by the encoding gene symbols throughout the paper.

**Imputation of GWAS summary statistics.** To maximise uniformity of all included GWASs (325 unique blood proteins and the migraine GWASs), Hap-Map3 SNPs were imputed using the Robust and Accurate Imputation from Summary Statistics (RAISS, https://statistical-genetics.pages.pasteur.fr/raiss/)[61]. We utilised 1000 G LD reference to approximate $Z$ scores of HapMap3 missing SNPs from neighbouring HapMap3 existing SNPs. The 1000 G LD matrices for Hap-Map3 SNPs were calculated for 1703 predefined LD-independent loci[62]. To uniform ambiguous SNPs (G/C and A/T) across all studied GWASs, we removed them before imputation and then imputed them using neighbouring SNPs. To estimate effect sizes from imputed $Z$ scores, standard errors were estimated using the reported sample size of the GWAS and 1000 G allele frequencies. We limited our imputation to HapMap3 common SNPs (MAF $\geq$ 0.01) and filtered imputed SNPs with $R^2 < 0.6$ to improve imputation quality. However, because imputation quality is correlated with LD scores[20], we performed LDSC and latent causal variable (LCV) analyses (where LD scores are used) on the original (not RAISS-imputed) GWAS summary statistics. All other genetic analyses were carried out on RAISS-imputed GWASs.

**Pleiotropy at genome-wide.** A significant correlation between $Z$ scores for LD-independent SNPs from migraine GWAS and a blood protein GWAS suggests a global concordant or discordant genetic effect called "genome-wide pleiotropy" in this paper. This approach is similar to the SNP effect concordant analysis (SECA) method[14] and has two main steps. The first step is to extract LD-independent SNPs from migraine (RIASS-imputed) GWAS, and the second step is to estimate the Pearson correlation ($r$) between LD-independent SNPs $Z$ scores for the migraine GWAS and the same set of SNPs from the blood protein (RIASS-imputed) GWASs. For extracting LD-independent SNPs, we applied $P$-value informed LD-clumping function of PLINK 1.9 (https://www.cog-genomics.org/plink2) on migraine RAISS-imputed GWAS using–clump-r2 0.1–clump-kb 10000 flags[63] and the 1000 G LD reference panel. This step identified 113,251 LD-independent SNPs with the smallest $P$-values at $r^2 < 0.1$ within their 10 Mb LD blocks. To minimise the multiple testing burden, the Pearson correlations were estimated between migraine and 270 (out of 325) blood proteins having > 60% overlap with the LD-independent migraine SNPs.

To deal with the multiple testing problem, FDR correction (Benjamini-Hochberg method) with an effective number of independent tests was used to adjust $P$-values. The spectral decomposition of a correlation matrix method (matSpD) was used to estimate the effective number of independent tests[64]. Briefly, the 270 blood proteins were estimated to be equivalent to 196.04 effective independent tests in the full dataset. All correlation results between $Z$ scores for LD-independent SNPs from the migraine GWAS and 270 blood protein GWASs are provided in Supplementary Data 2.

**Pleiotropy at 18,236 protein-coding genes.** Our study of pleiotropy at the gene level was comprised of two steps. In the first step, we performed gene-based analysis on RAISS-imputed GWASs for migraine and 325 blood proteins by applying multi-marker analysis of genomic annotation (MAGMA v1.07b, https://ctg.cncr.nl/software/magma). Our MAGMA analysis assigned SNPs to genes via an annotation window of ±500 kb flanking the gene boundaries derived from the matched Ensembl build (Ensembl build GRCh37) and 1000 G data, and then calculated $P$-values ($P_{gene}$) for each gene. We strictly considered a gene is associated with a trait when its Bonferroni-corrected gene-based $P$-value ($P_{gene}$) is $\leq$ 0.05 (i.e., $P_{gene} \leq 0.05/18,236$). The second step was to identify blood proteins that have pleiotropic effects with migraine at the gene level. We tested whether the observed proportion of genes associated with migraine in the subset of genes associated with a protein, is significantly greater than the null (expected) proportion of genes associated with migraine risk using the *Exact Binomial Test* (one-sided). The null proportion is 494/18,236, where 494 is the number of genes associated with migraine at Bonferroni-corrected $P_{gene} \leq 0.05$ from migraine GWAS and 18,236 is the number of all genes. The observed proportion for each protein is the number of genes associated with both migraine and the protein divided by the number of genes associated with the protein from the protein GWAS. This tests if genes associated with migraine risk are statistically enriched in the genes associated with blood protein levels. Estimated enrichment $P$-values

($P_{binomial-test}$) were corrected using the FDR (Benjamini-Hochberg method), identifying pleiotropy at the gene level between migraine and 15 blood proteins.

Furthermore, we conducted a combined analysis to identify pleiotropic genes and novel migraine-associated genes by increasing statistical power. We limited our combined analysis to the genes that pass the less strict Benjamini-Hochberg adjustment method in MAGMA results. Thus, $P_{gene}$ values from migraine GWAS were combined with $P_{gene}$ values from the 15 proteins if Benjamini-Hochberg adjusted $P_{gene}$ for migraine and one or more proteins are $\leq$ 0.05. We used Stouffer's Z method to combine $P_{gene}$ values. This method is more immune to very small $P_{gene}$ values (especially for blood proteins $P_{gene}$) than other existing combining methods such as the Fisher's method[65]. Moreover, to have an equal contribution between migraine and the 15 blood proteins for generating $Combined-P_{gene}$, we set weights one for $P_{gene}$ from each protein and equal to the number of included proteins (the number of proteins that have Benjamini-Hochberg adjusted $P_{gene} \leq 0.05$) for $P_{gene}$ from migraine. Therefore, small $P_{gene}$ from both migraine and proteins are required to yield a small $Combined-P_{gene}$. The original MAGMA result from migraine GWAS identified 494 genes associated with migraine risk at $P_{gene} \leq 0.05/18,236$ (Supplementary Fig. 1a, Supplementary Data 3) while the combined MAGMA results ($Combined-P_{gene}$ values) identified 651 genes associated with both migraine and at least one of the 15 identified proteins at $Combined-P_{gene} \leq 0.05/291,776$ that 291,776 is calculated based on the number of tests (16 × 18,236). Detailed results for combined analysis are available in Supplementary Data 4 and Supplementary Fig. 1b.

**Pleiotropy at SNPs within 1703 LD-independent loci.** To stud pleiotropy at SNPs within 1703 LD-independent loci, we applied pairwise GWAS (GWAS-PW, https://github.com/joepickrell/gwas-pw) on RAISS-imputed GWASs[17]. This analysis identifies overlapped genomic loci harbouring a shared SNP having an impact on both migraine risk and a blood protein level. GWAS-PW calculates four posterior probabilities (PPA) for 1703 predefined LD-independent loci[62] supporting four scenarios for each locus, [i] the association only to migraine (PPA1), [ii] the association only to the protein (PPA2), [iii] shared association to both migraine and the protein via the same SNP (PPA3), and [iv] shared association to migraine and the protein but via two distinct SNPs (PPA4). A PPA3 $\geq$ 0.9 was used to define a pleiotropic associated locus. We also calculated the proportion of loci affecting the protein level (the number of loci with PPA2 $\geq$ 0.9, PPA3 $\geq$ 0.9, or PPA4 $\geq$ 0.9) that also affecting migraine (the number of loci with PPA3 $\geq$ 0.9).

**Genetic correlation and causal relationship.** Genetic correlations ($r_g$) between migraine risk and blood levels of proteins were estimated using cross-trait LDSC[20]. This analysis was limited to the 58 blood proteins shown to have pleiotropy with migraine. The European pre-calculated LD scores from LDSC were used (similar to our heritability analysis). When the estimated cross-trait (genetic covariance) intercept was significant ($P$-value $\leq$ 0.05), the intercept was included to avoid the possible bias caused by unknown sample overlap between migraine and blood protein GWASs. However, for those blood proteins with non-significant genetic covariance intercept, we constrained the intercept to zero to reduce the standard error for $r_g$. Significant $r_g$ between migraine and blood proteins provides insight into the causation (vertical pleiotropy) or shared aetiology (horizontal pleiotropy) that can be distinguished by applying the latent causal variable (LCV) model[19]. As recommended by LCV developers, we limited our analysis to blood proteins that have a nominally significant $r_g$ ($P_{rg} < 0.05$) with migraine risk. Also, the significant $r_g$ provides critical information about the effect size and the sign of the significant causal relationships[19].

Therefore, we aimed to test for causal relationships between migraine as an outcome and only eight identified proteins (out of 58 blood proteins) that have a significant $r_g$ with migraine ($P_{rg} < 0.05$). The LCV model was carried out to estimate a genetic causality proportion (GCP) of blood proteins on migraine (https://github.com/lukejoconnor/LCV)[19]. A GCP of zero is interpreted as no genetic causality, while a GCP of one indicates complete genetic causality. The LCV model produces latent variables that explain a genetic correlation estimated between two traits. For example, if a protein has a genetic correlation with the latent variables stronger than the correlation with migraine, migraine is genetically caused by the protein.

**Pathway and tissue analyses.** Fictional enrichment tests were performed by g:Profiler/g:GOSt web-based tool (accessed Jun 2021, https://biit.cs.ut.ee/gprofiler/gost) using the default parameters[21].

Transcriptomic data were obtained from the consensus dataset generated by the Human Protein Atlas (HPA, version 20.1, https://www.proteinatlas.org/)[22]. This dataset provides RNA expression levels for 55 tissue types and 6 blood cell types as consensus normalised expression (NX) levels generated by combining data from HPA, GTEx and FANTOM5 (Supplementary Data 10).

Proteomic data were obtained from HPA[23]. This data provides protein expression levels for 55 tissue types comprising 105 cell types based on immunohistochemistry using tissue microarrays. The protein expression levels are reported in four categories "Not detected", "Low", "Medium" and "High" (Supplementary Data 11). For numerical analysis, we converted the protein

expression levels categories to numerical values as 0, 1, 2, 3 for "Not detected", "Low", "Medium" and "High", respectively.

The null hypothesis for the *t*-tests comparing the HPA RNA expression levels (NX levels) and protein expression levels (converted numerical values) of the 58 identified proteins to those proteins that have no detected pleiotropy with migraine (325−58 = 267) was that the means of two groups are equal and a two-sided *P*-value is reported ($P_{t\text{-test}}$).

**Reporting summary**. Further information on research design is available in the Nature Research Reporting Summary linked to this article.

## Data availability

The IHGC migraine GWAS 2016 data (except 23andMe, Inc. samples) are available for bona fide researchers, access can be obtained at http://www.headachegenetics.org/content/datasets-and-cohorts. The migraine GWAS summary statistics for the 23andMe discovery data set will be made available through 23andMe to qualified researchers under an agreement with 23andMe that protects the privacy of the 23andMe participants (https://research.23andme.com). All blood protein GWAS summary statistics are publicly available (Supplementary Table 1).

Transcriptomic and proteomic data were obtained from the consensus dataset generated by the Human Protein Atlas (HPA, version 20.1, https://www.proteinatlas.org/).

## Code availability

The Linux command-line scripts and R functions used to format GWAS summary statistics data and to perform statistical analyses are available from the corresponding authors upon request. Software and tools can be accessed for g:Profiler/g:GOSt web-based tool (accessed Jun 2021) at https://biit.cs.ut.ee/gprofiler/gost, for GWAS-PW (v0.21) at https://github.com/joepickrell/gwas-pw, for LCV (v1.0) at https://github.com/lukejoconnor/LCV, for LDSC (v1.0.1) at https://github.com/bulik/ldsc, for LocusZoom online tool (accessed Jun 2021) at http://csg.sph.umich.edu/locuszoom for MAGMA (v1.07b) at https://ctg.cncr.nl/software/magma, for PLINK (v1.9) at https://www.cog-genomics.org/plink2, for R (v4.0.3) at https://www.r-project.org/, and for RAISS (v1.0) at https://statistical-genetics.pages.pasteur.fr/raiss/.

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

## Acknowledgements

We thank the participants and many researchers involved in proteome GWAS studies. We thank the research participants and employees of 23andMe for making this work possible. H.M.T is grateful for support from the Queensland University of Technology through a QUT Postgraduate Research Scholarship.

## Author contributions

H.M.T. and D.R.N. conceptualised the study. H.M.T. collected and curated protein GWASs. H.M.T. carried out the formal analysis. H.M.T. wrote the initial draft. D.R.N. reviewed and edited the draft. D.R.N. supervised the work.

## Competing interests

The authors declare no competing interests.

## Additional information

# The International Headache Genetics Consortium

Dale R. Nyholt[2]

[2]School of Biomedical Sciences, Faculty of Health, Centre for Genomics and Personalised Health, Centre for Data Science, Queensland University of Technology (QUT), Brisbane, QLD, Australia.

A full list of members and their affiliations appears in the Supplementary Information.

