## [Peer Review File · Nature Communications]

Genetic analyses identify pleiotropy and causality for blood proteins and highlight Wnt/ β -catenin signalling in migraineReviewers' Comments:

Reviewer #2:

Remarks to the Author:

Tanha et al. evaluated shared genetic effect and/or causal relationships between blood proteins and migraine risk using GWAS summary statistics. They identified several blood proteins genetic related with risk of migraine, but most of these proteins were distinct from loci associated with migraine. They also identified potential causal relationships between multiple proteins and migraine risk. This is a well-designed and executed study and the manuscript is clearly written.

1. It is not clear and convincing that the analyses were limited to GWAS of phenotypes with nominally significant SNP heritability. The reasoning, which is based on their own unpublished study, that the genetic effect on proteome may be driven by monogenic signals rather than polygenic, is not supported by any data. Non-significant SNP heritability could be due to multiple factors, and it might be a missed opportunity to not examine pleiotropy and causal relationships of all proteins with migraine risk.
2. The authors refer multiple times to their own unpublished study. This is unusual – unless they can include the data in this study, they should not be making any inference from an unpublished study which has not undergone peer-review.
3. The number of proteins included in the cross-trait analysis was only 325 although they started with 4,625 blood protein GWAS studies. This is a significant drop. They should provide detailed and step-by-step exclusions justifying how they arrived at 325.
4. Was there any sample overlap between GWASs of multiple proteins and with those included in the migraine GWAS? Was it checked and accounted for if there was any?

Reviewer #3:

Remarks to the Author:

The authors use GWAS summary statistics from Migraine and 325 blood proteins to find horizontal pleiotropy and causality between blood proteins and migraine, finding 5 putative causal proteins.

- The authors calculate Z_{h2} and limit their analysis to significant proteins. They say that Z_{h2} is correlated to the size of the GWAS, therefore increasing GWAS size may allow them to increase the number of proteins. But they also say that when there was a repeated trait they took the most significant one. Why didn't the authors try to do a meta-analysis for repeated traits? Is it because there may be an overlap between the proteomics datasets?

-To calculate the correlation between migraine and a blood protein GWAS they just calculated the correlation between z-scores. Techniques as LD Score would be a better way to calculate if there is a correlation of the underlying genetic effects.

-In the identification of genes with effect to both migraine and blood protein levels: "Next, to identify blood proteins that have pleiotropic effects with migraine at the genes level, we tested whether the proportion of genes associated with migraine in the subset of genes associated with a protein, is significantly greater than the null (expected) proportion of genes associated with migraine risk using the Exact Binomial Test."

In this case I would expect the genes associated to a trait to be correlated rather than to be a random sampling. Therefore, it would be better to sample genes with the same correlation distribution and see if the proportion of genes is greater rather than assuming that gene expression is randomly distributed.

minor comments:

-In the description of the cohorts, in population the study Suhre 2017 (1,335 / 1,124) – Plasma proteome

- Population: KORA

It is not clear what KORA means. Is it also individuals from European ancestry?

Repeated migraine risk in discussion: "Moreover, we found genome-wide pleiotropy between migraine risk and lower levels of an accessory subunit of voltage-gated potassium channels, KCNE2, and migraine risk"

Reviewer #2 (Remarks to the Author):

Tanha et al. evaluated shared genetic effect and/or causal relationships between blood proteins and migraine risk using GWAS summary statistics. They identified several blood proteins genetic related with risk of migraine, but most of these proteins were distinct from loci associated with migraine. They also identified potential causal relationships between multiple proteins and migraine risk. This is a well-designed and executed study and the manuscript is clearly written.

1. It is not clear and convincing that the analyses were limited to GWAS of phenotypes with nominally significant SNP heritability. The reasoning, which is based on their own unpublished study, that the genetic effect on proteome may be driven by monogenic signals rather than polygenic, is not supported by any data. Non-significant SNP heritability could be due to multiple factors, and it might be a missed opportunity to not examine pleiotropy and causal relationships of all proteins with migraine risk.

Response: We appreciate the reviewer's point regarding not including all 4,625 blood proteins GWASs in our analyses. However, as migraine, like many other common complex traits, is highly "polygenic", we aimed to study shared "polygenic" genetic effects (relationship) between migraine and blood proteins. Thus, we only focused on 325 unique blood proteins GWASs with a significant polygenic h_{SNP}^2 (estimated by LDSC) to have any statistical power for exploring "polygenic" pleiotropy—i.e., to detect significant genetic correlation (bivariate heritability) between two traits, each trait must itself have significant (univariate) heritability. Filtering out blood proteins with non-significant polygenic h_{SNP}^2 also minimises the multiple testing burden. Hence, we believe that this step is essential and consistent with our aim. Nevertheless, in the revised manuscript, we have added a paragraph listing potential limitations of this study where we suggested that other studies may investigate the shared genetic effects between blood proteome and migraine under the local heritability model (*cis*-pQTL model). To reduce confusion, we now better define and use "local heritability" terminology instead of "monogenic" in the revised manuscript.

In the section "**4.1 Blood proteins with significant polygenic signals**", we compared the polygenic h_{SNP}^2 results for 4,625 blood **proteins** (from this study) to the polygenic h_{SNP}^2 results for 972 blood **metabolites** from our previous study that has been published now (Tanha HM et al.; Am J Hum Genet. 2021 Nov; PMID: 34644541) (1). For example, we observed that Z scores of h_{SNP}^2 ($Z_{h^2_{SNP}}$) from blood proteins are not correlated with the number of included SNPs in LDSC analysis ($P = 0.87$). Conversely, estimated $Z_{h^2_{SNP}}$ for 972 blood metabolites is significantly associated with the number of included SNPs (1), suggesting a lower (genome-wide) polygenicity for blood proteome than blood metabolome. Also, we found that only 7.8% (362/4,625) of blood proteins showed significant h_{SNP}^2 , compared to blood metabolome where 41.67% (405/972) of metabolites showed significant h_{SNP}^2 (1). This again suggests that the levels of blood proteins are more attributable to local heritability than polygenicity (global heritability) where many *trans*-pQTLs are involved.

Proteins and metabolites are both considered intermediate molecular phenotypes connecting the genome to disease endpoints. Thus, we believe that these comparisons between proteins and metabolites h_{SNP}^2 are interesting and informative for future genetic studies. However, these comparisons are in addition to our primary aim of investigating polygenic pleiotropy between blood proteins and migraine; thus, not affecting our workflow or conclusion. Nonetheless, should the reviewers and editor prefer, we are happy to remove the relevant text from our manuscript.

2. The authors refer multiple times to their own unpublished study. This is unusual – unless they can include the data in this study, they should not be making any inference from an unpublished study which has not undergone peer-review.

Response: We appreciate this read/appeared unusual and should normally be referred to as ‘unpublished data’, etc.—the timing was awkward regarding our current submission (to *Nat Commun*) and acceptance of our earlier study. We’re very happy to note that our previous study on blood metabolome and migraine is now published (Tanha HM et al.; *Am J Hum Genet.* 2021 Nov ;PMID: 34644541). We also have updated our citation for this study in the revised manuscript.

In addition to our primary aim, we made some comparisons between polygenic heritability results from blood proteome and blood metabolome. From **Discussion** in the revised manuscript, “Comparing the heritability results from the present study to our recent study of blood metabolome (1) suggests a lower polygenicity for blood proteome than blood metabolome. First, polygenic SNP heritability was not associated with the number of included SNPs for the 4,625 blood proteins, whereas it was significantly associated in our study of 972 blood metabolites. Second, a smaller proportion of blood proteins showed significant polygenic h_{SNP}^2 compared to blood metabolites (7.8% versus 41.67%). Taken together, our comparisons suggest that blood metabolites appear more polygenic than blood proteins, which is plausible given multiple proteins can play catalytic or regulatory roles in anabolism, catabolism, absorption, secretion and transportation of a metabolite (2).”

Please note that these comparisons do not affect our primary aim of investigating polygenic pleiotropy and causality between blood proteins and migraine. More clarifications have been added in the revised manuscript.

3. The number of proteins included in the cross-trait analysis was only 325 although they started with 4,625 blood protein GWAS studies. This is a significant drop. They should provide detailed and step-by-step exclusions justifying how they arrived at 325.

Response: Thank you for this suggestion. To add more clarification, we have added a comprehensive step-by-step legend for our workflow figure (**Figure 1**). Also, detailed information for inclusion/exclusion criteria is available in section “**6.2 Estimation of SNP heritability and filtration**” in the revised manuscript. We also note that we now better describe why only the 325 blood protein GWAS with significant *univariate* heritability were used (i.e., only these GWAS have potential to produce a significant *bivariate* heritability)

4. Was there any sample overlap between GWASs of multiple proteins and with those included in the migraine GWAS? Was it checked and accounted for if there was any?

Response: We appreciate the reviewer's concern about possible sample overlap between migraine GWAS and proteins GWAS. There is no sample overlap that we are aware of. Unknown sample overlap between a small number of cohorts within the migraine GWAS meta-analysis and protein GWAS meta-analyses may exist; however, compared to the total number samples in the large GWAS meta-analyses, such overlap would be negligible. Moreover, given the protein GWASs are quantitative traits, they are not expected to suffer the potential bias of performing cross-trait genetic analyses of binary (e.g., affection) traits that is driven by genetic similarity across the control samples (and/or case samples if highly comorbid).

Also, the genetic covariance intercept from bivariate LDSC analysis can estimate sample overlap between two GWASs, where genetic covariance intercept of zero means no sample overlap and genetic covariance intercept of one means perfect sample overlap. The estimated genetic covariance intercept between migraine GWAS and the 325 analysed protein GWASs are mostly not significantly different from zero. In section "**6.7 Genetic correlation and causal relationship**", we explained that "when the estimated cross-trait (genetic covariance) intercept was significant (P -value ≤ 0.05), the intercept was included to avoid the possible bias caused by unknown sample overlap between migraine and blood protein GWASs. However, for those blood proteins with non-significant genetic covariance intercept, we constrained the intercept to zero to reduce the standard error for genetic correlation (r_g)." For our causality analysis by the LCV model, we only included proteins having a significant LDSC genetic correlation with migraine adjusted for unknown sample overlaps. Despite being confident that our findings are robust to any potential small overlap in samples, we now discuss this potential limitation in the revised manuscript.

Reviewer #3 (Remarks to the Author):

The authors use GWAS summary statistics from Migraine and 325 blood proteins to find horizontal pleiotropy and causality between blood proteins and migraine, finding 5 putative causal proteins.

5. The authors calculate Z_{h^2} and limit their analysis to significant proteins. They say that Z_{h^2} is correlated to the size of the GWAS, therefore increasing GWAS size may allow them to increase the number of proteins. But they also say that when there was a repeated trait they took the most significant one. Why didn't the authors try to do a meta-analysis for repeated traits? Is it because there may be an overlap between the proteomics datasets?

Response: Yes, results from our heritability analysis showed that increasing sample size for proteins GWASs will produce larger $Z_{h^2_{SNP}}$. Although meta-analysis is a good idea, as the reviewer pondered, the larger protein GWASs are already meta-analyses of previous protein GWASs; thus, we are not able to easily perform meta-analysis due to sample overlap between blood protein GWASs. For example, proteins GWASs from Folkersen 2020 study (Folkersen L et al.; Nat Metab. 2020 Oct; PMID: 33067605) are meta-analyses results of Folkersen 2017 (Folkersen L et al.; PLoS Genet. 2017 Oct; PMID: 28369058) and additional cohorts.

6. To calculate the correlation between migraine and a blood protein GWAS they just calculated the correlation between z-scores. Techniques as LDSC would be a better way to calculate if there is a correlation of the underlying genetic effects.

Response: We appreciate the reviewer's suggestion and are aware of the excellent utility of LDSC genetic correlation; indeed, we report LDSC genetic correlation results in our manuscript. However, in the present study, we aimed to explore pleiotropy (rather than genetic correlation) between blood proteins and migraine at three different levels (genome-wide, gene, and SNP). For genome-wide pleiotropy, we estimated correlation between LD-independent SNP Z-scores that robustly tests whether genetic factors (global SNP allele effects) influencing blood proteins levels are significantly concordant or discordant with genetic influences underlying migraine risk. Briefly, this approach provides the direction of the relationship (like LDSC genetic correlation) and is intuitive and effective to identify *significant* pleiotropy at the genome-wide level, and importantly is less sensitive to the protein GWAS sample size. However, this approach is not well-suited to measuring the *magnitude* of the relationship which LDSC genetic correlation is able to estimate. We discussed this in the added limitation paragraph in the revised manuscript.

Notably, for our causality analysis by the LCV model, we restricted our test to only proteins having a significant LDSC genetic correlation with migraine (**Table 2**) that is also recommended by the LCV model developers (O'Connor LJ, Price AL; Nat Genet. 2018 Nov; PMID: 30374074).

7. In the identification of genes with effect to both migraine and blood protein levels: "Next, to identify blood proteins that have pleiotropic effects with migraine at the genes level, we tested whether the proportion of genes associated with migraine in the subset of genes associated with a protein, is significantly greater than the null (expected) proportion of genes associated with migraine risk using the Exact Binomial Test."

In this case I would expect the genes associated to a trait to be correlated rather than to be a random sampling. Therefore, it would be better to sample genes with the same correlation distribution and see if the proportion of genes is greater rather than assuming that gene expression is randomly distributed.

Response: We would like to thank the respected reviewer for raising this point. Here we performed our analysis comparable to previous study of pleiotropy (Watanabe K; Nat Genet. 2019 Sep; PMID: 31427789) where the number of uncorrelated genes was not considered for estimating the percentage of the genes that show pleiotropy between complex traits. Similarly, in another study (Jones SE; Nat Commun. 2019 Jan; PMID: 30696823), similar approach was used. The less strict assumption for these studies and ours is that the rate of unwanted biases (such as correlation between genes) in the null and observation proportions is similar.

However, we agree with the respected reviewer that the correlation between co-located genes across a GWAS locus could produce smaller p-values in the binomial test. Hence, we assessed the inflation of the test statistics as a result of correlation between genes by generating a quantile-quantile (Q-Q) plot of observed-versus-expected $-\log_{10}$ p-values. The Q-Q plot does not indicate an excess of smaller p-values, hence we believe our pleiotropy analyses (binomial test) to be robust.

minor comments:

8. In the description of the cohorts, in population the study Suhre 2017 (1,335 / 1,124) – Plasma proteome

- Population: KORA

It is not clear what KORA means. Is it also individuals from European ancestry?

Response: Thank you for requesting this clarification. KORA is the general population living in the region of Augsburg, southern Germany. Detailed information related to all include proteome GWAS studies (for 4,625 blood proteins), including population of interest have been provided in the new **Supplementary Table 1.**

9. Repeated migraine risk in discussion: "Moreover, we found genome-wide pleiotropy between migraine risk and lower levels of an accessory subunit of voltage-gated potassium channels, KCNE2, and migraine risk"

Response: Thank you for noting this. It is corrected now.

Reference:

1. Tanha HM, Sathyanarayanan A, Nyholt DR. Genetic overlap and causality between blood metabolites and migraine. *Am J Hum Genet.* 2021;108(11):2086-98.
2. Nielsen J. Systems Biology of Metabolism: A Driver for Developing Personalized and Precision Medicine. *Cell Metab.* 2017;25(3):572-9.

Reviewers' Comments:

Reviewer #2:

Remarks to the Author:

The authors have addressed my previous concerns. No further comments!

Reviewer #3:

Remarks to the Author:

Thanks very much to the authors for addressing my concerns. I am happy with the manuscript now.